# Potential Impacts of Climate Change on Surface Water Resources in Arid Regions Using Downscaled Regional Circulation Model and Soil Water Assessment Tool, a Case Study of Amman-Zerqa Basin, Jordan

**Ibrahim Al-Hasani** [1] , **Mohammed Al-Qinna** [1] **and Nezar Atalla Hammouri** [2,3,]*

[1] Department of Land Management and Environment, Prince El-Hassan Bin Talal Faculty for Natural Resources & Environment, The Hashemite University, Zarqa 13133, Jordan

[2] Department of Applied Physics and Astronomy, Faculty of Science, University of Sharjah, Sharjah 27272, United Arab Emirates

[3] Department of Earth and Environmental Sciences, Prince El-Hassan Bin Talal Faculty for Natural Resources & Environment, The Hashemite University, Zarqa 13133, Jordan

* Correspondence: nhammouri@sharjah.ac.ae

**Abstract:** Water scarcity, aggravated by climate change impacts, threatens all sectors in arid regions and hampers sustainable development plans. This work aims to assess the potential impacts of climate change on surface water resources of Amman-Zerqa Basin, Jordan, using the Soil Water Assessment Tool model (SWAT) and outputs from the Downscaled Regional Circulation Model. Future scenarios were developed based on combining two Representative Concentration Pathways (RCPs 4.5 and 8.5). A reference scenario from 1973 to 2015 was used to compare the current climate with future climates and their impacts on hydrological processes. Hydrologic modeling outputs showed very good performance ratings for calibration and validation periods. Statistical bias correction of the Downscale Regional Circulation Model (GCM) indicated that linear scaling for precipitation data was the best-performing bias correction method, along with variance scaling and distribution mapping methods for minimum and maximum temperature, respectively. The coupled future model simulations indicated a reduction in crucial water balance components under all modeled scenarios. The simulated reductions range between 3.7% and 20.7% for precipitation, 22.3–41.6% for stream flow, 25.0–47.0% for surface runoff, 0.5–13.4% for evapotranspiration, and 21.5–41.4% for water yield, from conservative to the severe scenario, respectively. In conclusion, spatial analyses indicated the presence of three zones of impact. Thus, future climate and hydrological adaptation measures should focus on the provided zoning.

**Keywords:** hydrologic modeling; soil and water assessment tool; climate change; downscaled regional circulation model; arid regions; Jordan

## 1. Introduction

Human-induced climate change poses a global challenge to the natural and human systems, threatening sustainable development and aggravating risks for poor and less developed nations. The persistent greenhouse gas (GHGs) emissions will cause further global warming and long-term changes in all components of the climate system, aggravating the likelihood of severe, pervasive, and irreversible impacts on people and ecosystems [1]. Climate projections for precipitation suggest non-uniform changes globally, where the mid-latitude and subtropical dry regions will likely face decreasing mean precipitation. This will intensify current risks and generate new risks for natural and human systems [2].

Future climate simulations for the Mediterranean region and specifically in arid and semi-arid regions suggest a noticeable increase in air temperature and a decrease in annual precipitation by the end of the century attributed to increased anticyclonic circulation



resulting in increasingly stable conditions [2]. The latest climate change assessment in Jordan was achieved using the Coordinated Regional Climate Downscaling Experiment (CORDEX) derived from the Coupled Model Intercomparison Project Phase-5 (CMIP5) of the World Climate Research Programme (MOE and UNDP, 2014). Future climatic projections suggest extreme mean temperature rise to 2.1 °C (1.7 to 3.1 °C) for Representative Concentration Pathway (RCP) of 4.5 and 4 °C (3.8 to 5.1 °C) for RCP 8.5 with warmer summers. Additionally, the models suggest a likely decrease of cumulated precipitation by 15% (6% to 25%) in RCP 4.5, and 21% (9% to 35%) in RCP 8.5 between 2070 to 2100 [3].

The assessment of climate change's impact on the hydrological systems depends mainly on the coupling of climate and hydrological models' simulations, where climate models are used to generate weather variables required as input for hydrological models to represent the hydrological systems under study [4]. In general, climate models are simplified, computerized mathematical representation of the earth's climate system that characterizes the energy budget and various climatic feedback mechanisms, which control the climate at the global and regional scales [4,5]. Climate models can be categorized from simple to complex models as One-dimensional Energy Balance Models (EBMs), One-dimensional radiative-convective models, Two-dimensional Statistical-Dynamical Models (SDMs), and Three-dimensional General circulation models (GCMs) [4,5].

Since the GCMs results are very coarse in resolution, they are only considered for global and continental scale studies (>100 km) requiring mean climate variables at long time scales (>monthly). Therefore, dynamic and statistic Downscaled Regional Climate Models' (RSMs) outputs are preferred for hydrological studies tackling climate change issues to address future changes at the national, local, and basin scales [4]. GCM is incorporated with land/sea features and physical processes to better represent the climate system at high spatial resolutions (20–50 km) [6].

The second essential component of climate change impact assessment is the hydrologic model, that is to say, a simplified mathematical representation developed to study the dynamic processes of the hydrologic systems and predict their behaviors for various water resource management purposes [4]. Selecting a suitable model is essential and requires a sufficient understanding of model types, functionalities, applicability, and availability [7]. In general, simple empirical water balance or rainfall-runoff models (e.g., Artificial neural network, unit hydrograph) are suitable for studying climate variables change on the basin hydrology on a regional scale, lumped-parameter conceptual models (e.g., HBV model, TOPMODEL) are suitable for quantifying surface flow, and physically based distributed models (e.g., SHE or MIKESHE model, SWAT) are suitable for studying spatial patterns of hydrologic properties within watersheds [4,8].

The SWAT model is a physically based, semi-distributed watershed hydrologic model jointly developed by the United States Department of Agriculture (USDA)—Agricultural Research Service (ARS), and Texas A&M AgriLife Research, part of The Texas A&M University System. SWAT is a small watershed-to-river basin-scale model that simulates the quality and quantity of surface and groundwater and predicts the environmental impact of land use, land management practices, and climate change [9]. In SWAT, hydrologic processes such as surface runoff, infiltration, evapotranspiration, canopy interception, and soil water movement are simulated at the Hydrologic Response Units (HRU) level, each one separately, assuming no interaction between HRUs in one subbasin [9].

Given the vulnerability of the scarce and stressed water resources to climate change in arid regions, it is essential to quantify potential climate change impacts on its water balance components both temporally and spatially to inform climate actions and decision-making processes fostering integrated water management and resilience. This article aims to estimate the future potential impacts of climate change on surface water resources by integrating downscaled output from a Regional Circulation Model with Soil Water Assessment Tool in an arid region. The key outputs of this investigation show the types and magnitudes of impacts of downscaled climate change projections from the Regional

Circulation Model on water balance components, using a semi-distributed hydrologic model (SWAT), in a mixed land-use arid AZB.

In this study, we constructed and parameterized SWAT hydrological model to reflect the key water balance components of AZB, then we Downscaled Regional Circulation Model outputs and tested their performance in reproducing historically observed weather variables using bias correction approaches, then derived future bias-corrected climate projections in terms of precipitation and temperature change, and finally simulated the potential impacts of climate change on the water balance component of AZB spatially and temporally through forcing the calibrated SWAT model with bias-corrected downscaled RCM future climate scenarios. To the best of the authors' knowledge, this study is believed to be the most up-to-date and detailed analysis of climate change impacts on surface water balance components of AZB, using state-of-the-art scientific approaches and updated data.

## 2. Materials and Methods

### 2.1. Study Area

As in many arid regions, Jordan has limited and stressed water resources due to anthropogenic and naturogenic factors. The most crucial river basin system in Jordan due to its geographic, demographic, socio-economic, and above all, the current water resources value for the country is the Amman-Zarqa surface water basin (AZB), one of the largest watersheds in the country. The basin covers an area of about 4120 km$^2$, extending in the northeast from Jebal al-Arab in Syria at an elevation of 1460 m above seas level (asl) to its outlet in the west near Dair Alla at an elevation of 200 m below sea level (bsl). The Syrian part of the basin (about 5% of the basin area) was excluded from this study due to the lack of available data. The AZB is located in north Jordan between latitudes 31.86–32.5° North and longitudes 35.67–36.7° East (Figure 1). The basin hosts the capital city Amman, in addition to other major cities such as Zarqa and Jerash, where about 65% of the country's population lives (The Jordanian Department of Statistics [DOS], 2017), holding about 80% of its industries, and contributing to major agricultural activities. The surface water resources of the Zarqa River are used principally for irrigation and stock watering; due to its sources and quality, it is a strategic resource for water in Jordan.

The AZB is characterized by an eastern Mediterranean semi-arid climate with cold, humid winter and hot, dry summer, with two short transitional periods in autumn and spring. The mean annual rainfall ranges from less than 100 mm in the arid desert in the east to more than 500 mm in the forest highlands in the west. The annual average temperature in the basin is 17.3 °C. The daily average temperature ranges from approximately 8 °C in winter to 25 °C in summer. The monthly maximum and minimum temperatures range between 13.7–3.5 °C in January and 34.7–18.8 °C in July [10]. The basin received about 666.2 MCM of rainfall in the 2013/2014 season, of which 592.9 MCM goes as evaporation (90%), 26.7 MCM as runoff (4%), and 46.6 MCM as infiltration (7%). Nonetheless, these amounts differ yearly and from wet to dry periods [11]. The derived water budget for AZB over thirty years reported 273 MCM of weighted average rainfall in a year (1998/1999) and 1343 MCM for a wet year (1991/1992) [10].

The main land uses/land cover types for the AZB are classified into eight main classes (Figure 2a): rangelands composed of sands, chert plains, dry and wet mudflats, Wadi deposits, and bare soil (RANGE = 43%) of the basin area, followed by pastures (PAST = 16.8%), agricultural lands cultivated with vegetables and field crops (AGRL = 10.8%), barren lands of basaltic and bare rocks, and quarries (BARR = 9.5%), orchards of tree crops (ORCD = 9%), urban fabric including cities such as Amman, Zarqa, Jerash, and Mafraq (URBN = 6%), mixed forests of open and closed trees (FRST = 4.7%), and some areas of water bodies such as dams (KTD) and wastewater plants (As-Samra WWTP) (WATR = 0.17%), respectively.

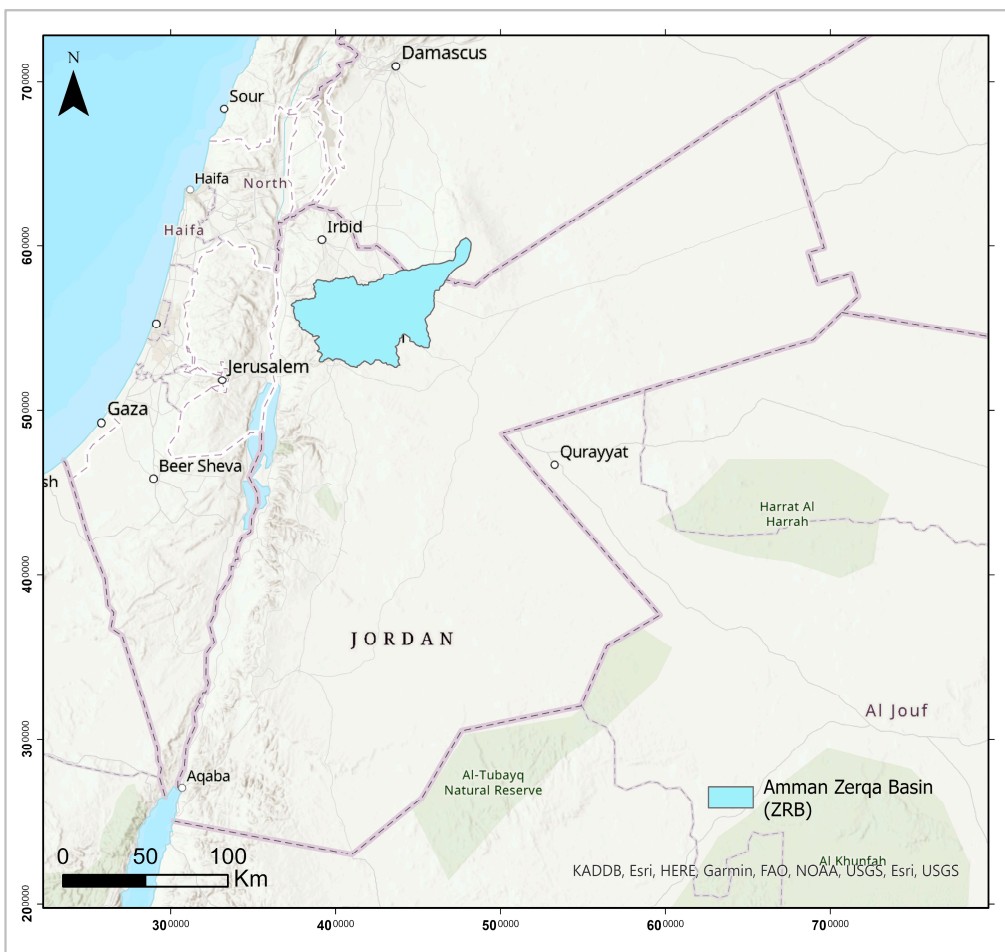

**Figure 1.** Location of the Amman-Zarqa Basin.

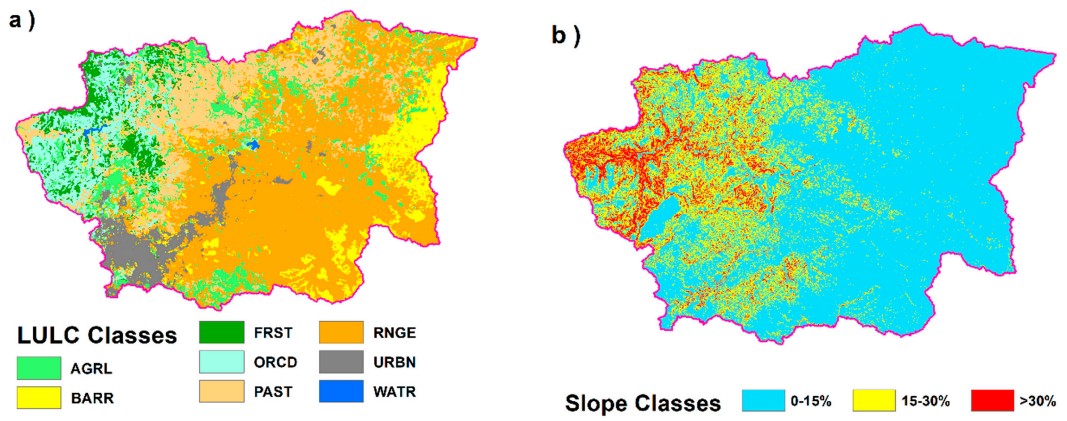

**Figure 2.** *Cont.*

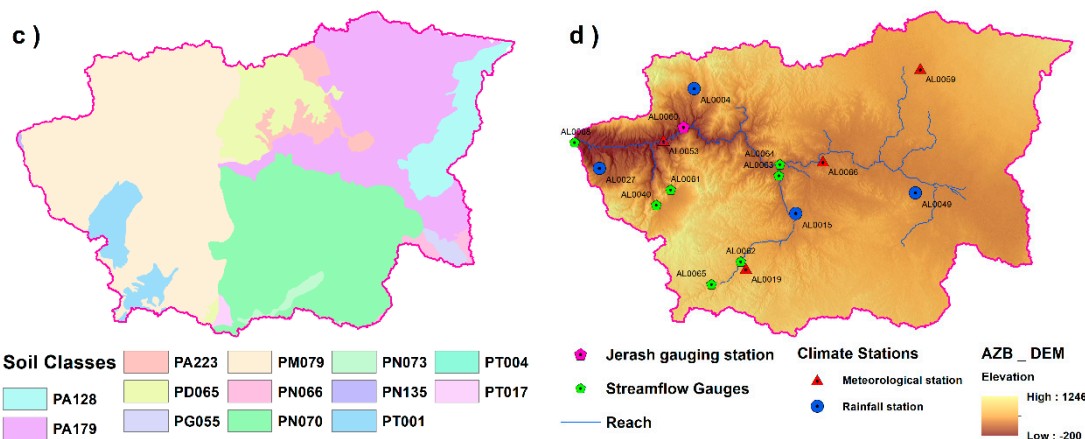

**Figure 2.** GIS-based parameters used in SWAT modeling, (**a**) Land use/Land cover (LULC) classes, (**b**) Slope Classes, (**c**) Soil classes and (**d**) Main streams and climate stations.

### 2.2. SWAT Modeling

ArcSWAT version 2012_10_3.19 interface for ArcGIS 10.1 was used to simulate the historical and future projected water hydrology of the AZB. Three spatial input datasets were prepared for SWAT modeling, including the Digital Elevation Model, Land use/Land cover, and soil. Varied information database files for different aspects of the watershed climate and hydrology, including lookup tables, gauging station's location, and daily weather data, were provided [9,12]. Table 1 summarizes the different data sets used in this study to develop the SWAT model.

**Table 1.** Data sets used to develop SWAT Model.

| Data Set | Details | Source | Figure |
|---|---|---|---|
| Land use/land cover | USGS LULC classifications level 1 | Ministry of Water and Irrigation, Jordan | Figure 2a |
| Topographic Data | Shuttle Radar Topography Mission (SRTM)-30 m | USGS Earth Explorer | Figure 2b |
| Soil Map Units | Level 1 National Soil Map and Land-use Project of Jordan | Ministry of Agriculture | Figure 2c |
| Weather Dataset | Precipitation, maximum and minimum air temperature, solar radiation, wind speed, and relative humidity | Ministry of Water and Irrigation, Jordan | Figure 2d |
| Streamflow | Observed streamflow data | Ministry of Water and Irrigation, Jordan | Figure 2d |

### 2.3. Development of Reference Scenario Using SWAT

Weather station locations were loaded and assigned to the sub-watersheds by linking each subbasin to one gage. The hydrological cycle in the land phase was simulated using the SWAT model based on the following water balance equation [9].

$$SW_t = SW_o + \sum_{i=1}^{t} \left( R_{day} - Q_{surf} - E_a - w_{seep} - Q_{gw} \right) \tag{1}$$

where $SW_t$ is the final soil water content (mm H2O), $SW_0$ is the initial soil water content on the day $i$ (mm H2O), $t$ is the time (days), $R_{day}$ is the amount of precipitation on day $i$ (mm H2O), $Q_{surf}$ is the amount of surface runoff on day $i$ (mm H2O), $E_a$ is the amount of evapotranspiration on day $i$ (mm H2O), $w_{seep}$ is the amount of water entering the vadose zone from the soil profile on day $i$ (mm H2O), and $Q_{gw}$ is the amount of return flow on day $i$ (mm H2O).

The SWAT simulation was run from 1 January 1970 to 31 December 2015 monthly, using three years as a warm-up period. The skewed normal rainfall distribution was selected. The model run results were saved and checked using the SWAT error checker that reads model output from a SWAT project and performs many simple checks to identify potential model problems, especially the hydrologic balance components.

The Penman-Monteith method was used to estimate the potential evapotranspiration [13]. The rainfall-runoff calculation was estimated using the SCS curve number method [14] and the channel routing calculation was estimated using the Variable Storage Routing method [15].

Manual calibration was achieved through the regionalization of parameters for a single streamflow gauging station (Jerash Bridge AL0060) [16].

In order to have a reasonable surface water balance in all hydrologic processes according to published resources [10,17], and to have a consistent rainfall-runoff hydrograph response to observed and simulated streamflow plots, the default parameters were modified based on the best knowledge of the surface water regime, one at a time, on a stepwise parameter modification until the satisfactory model performance was obtained [18]. The results were tested using the simple statistics of the Coefficient of Determination ($R^2$) and the root-mean-square error ($RMSE$), as follows:

$$R^2 = \frac{\left[\sum_i^n \left(Qi - \overline{Q}m\right)\left(\hat{Q}i - \overline{Q}s\right)\right]^2}{\sum_i^n \left(Qi - \overline{Q}m\right)^2 \sum_i^n \left(\hat{Q}i - \overline{Q}s\right)^2} \tag{2}$$

$$RMSE = \frac{\sqrt[2]{\sum_i^n \left(Qi - \hat{Q}\right)^2}}{n} \tag{3}$$

where $Qi$ is the measured discharge on day $i$, $\hat{Q}$ is the simulated discharge on day $i$, $\overline{Q}m$ is the mean measured discharge, and $\overline{Q}s$ is the mean simulated discharge.

Automated calibration, validation, sensitivity, and uncertainty analysis were conducted using the SWAT-CUP software package through the Sequential Uncertainty Fitting version 2 (SUFI-2) optimization algorithm [18]. This iterative algorithm process attempts to hold most of the measured data within 95% prediction uncertainty (95PPU), representing 2.5% and 97.5% levels of the cumulative distribution of an output variable obtained through Latin hypercube sampling [18]. This method proved to have an advantage over others in model calibration and uncertainty analysis as it could be run with the smallest simulation runs to achieve good prediction uncertainty bands and model performance [19].

The measured data were divided into two independent datasets; the calibration period dataset was covering the period from 1973 to 2002, and the validation dataset for the rest of the measurement history from 2003 to 2015.

In order to identify the most important influencing factor in the SWAT model that represents vital hydrological processes in the study basin and aims at decreasing the number of parameters in the calibration procedure, 16 selected parameters were tested using one iteration of 200 simulations (Table 2). The selection was guided through previous SWAT analyses and reviews [16,20].

**Table 2.** The initial list of parameters used for sensitivity analysis.

| Parameter | Input File Type | Description |
|---|---|---|
| ESCO | Basin | Soil evaporation compensation factor |
| SURLAG | Basin | Surface runoff lag coefficient |
| EPCO | Basin | Plant uptake compensation factor |
| SOL_K | Soil | Saturated hydraulic conductivity (mm/hr) |
| SOL_AWC | Soil | Available water capacity of the soil layer ((mm H2O/mm Soil) |
| SOL_BD | Soil | Moist bulk density (g/cm$^3$) |
| SOL_ALB | Soil | Moist soil albedo (%) |
| GW_REVAP | Groundwater | Groundwater "revap" coefficient |
| REVAPMN | Groundwater | Threshold water depth in the shallow aquifer for "revap" to occur (mm) |
| ALPHA_BF | Groundwater | Baseflow alpha factor (days) |
| GW_DELAY | Groundwater | Groundwater delay (days) |

**Table 2.** *Cont.*

| Parameter | Input File Type | Description |
| --- | --- | --- |
| GWQMN | Groundwater | Threshold depth of water in the shallow aquifer required for return flow to occur (mm) |
| CANMX | HRU | Maximum canopy storage (mm H2O) |
| OV_N | HRU | Manning's "n" value for overland flow |
| CH_K2 | Main channel | Effective hydraulic conductivity in main channel alluvium (mm/h) |
| CN2 | Management | SCS runoff curve number for moisture condition II |

Two types of parameter sensitivity analysis were used to parameterize the final model [18]; one at a time (OAT) or local sensitivity analysis was conducted using five simulations for each selected parameter, in which all parameters are held constant while changing one targeted parameter to identify its effect on streamflow and objective functions, and all at a time (AAT) or global sensitivity analysis, where all parameters are changing in each simulation run. Hence, 300 simulation runs using four iterations were conducted to identify its effect on streamflow and objective functions. The AAT uses a multiple regression approach to quantify the sensitivity of each parameter, where a t-test is used to identify the relative significance of each parameter; the larger in absolute value the t-stat, and the smaller the *p*-value, the more sensitive the parameter is.

Two indices are used in SUFI-2 to express the goodness of fit between model simulation (expressed as 95% prediction uncertainty '95PPU') and observed data (measured signal plus error); The P-factor is the percentage of observed data within the 95PPU band and varies from 0 to 1, where 1 indicates 100% bracketing of the measured data within model prediction uncertainty. The R-factor represents the thickness of the 95PPU range and is expressed as the ratio of the average width of the 95PPU band and the standard deviation of the measured variable; it varies from zero to infinity, where closer values to zero are more desirable [21].

The strength of the calibration and validation outcomes is judged based on those two indices. A larger P-factor can be achieved at the expense of a larger R-factor. Hence, a balance must be reached between the two indices. A value of >70% for P-factor and R-factor <1.5 for discharge modeling is considered satisfactory, but this is subjective and varies according to the study conditions and adequacy of the input and calibration data [21,22].

Four iterations were run for the calibration period (1973–2002) to reach acceptable values of the R-factor and P-factor. In the final iteration, the new parameter ranges were used as calibrated parameters for further modeling. SUFI-2 provides ten objective functions to be employed as targets during the iterative calibration [21]. The three most common performance indices (objective functions) were used to assess the performance of the SWAT model as follows:

1.　Nash-Sutcliffe efficiency (NSE): determines the relative magnitude of the residual variance compared to the measured data variance [23]. Its value ranges from $-\infty$ to 1, where 1 indicates a perfect model and a value less than 0 indicates that the mean value of the observed time series would have been a better predictor than the model.

$$\text{NSE} = 1 - \frac{\sum_{i=1}^{n}\left(Qi - \hat{Q}i\right)^2}{\sum_{i=1}^{n}\left(Qi - \overline{Q}\right)^2} \tag{4}$$

where $Qi$ is the measured discharge on day $i$, $\hat{Q}$ is the simulated discharge on day $i$, and $\overline{Q}$ is the mean measured discharge.

2.　Percent bias (PBIAS): measures the difference between the simulated and observed quantity, and its optimum value is 0. A positive value of the model represents underestimation, whereas a negative value represents the model overestimation [23].

$$\text{PBIAS} = 100 * \frac{\sum_{i=1}^{n}\left(Qi - \hat{Q}i\right)}{\sum_{i=1}^{n} Qi} \tag{5}$$

3. The ratio of the root-mean-square error to the standard deviation of measured data (RSR): a complementary indicator to RMSE, it standardizes the RMSE using the observation standard deviation. The optimum value of RSR is 0 and a higher value indicates lower model performance [23].

$$\text{RSR} = \frac{\sqrt[2]{\sum_{i=1}^{n}\left(Qi - \hat{Q}i\right)^2}}{\sqrt[2]{\sum_{i=1}^{n}\left(Qi - \overline{Q}\right)^2}} \tag{6}$$

In general, model simulation can be judged as satisfactory if NSE > 0.50 and RSR $\leq$ 0.70, and if PBIAS $\pm$ 25% for streamflow [23]. To build confidence in the calibrated model parameters, validation was performed by applying the calibrated parameter ranges on an independently measured streamflow dataset from 2003 to 2015. The validation process requires one iteration using the same number of simulations run (i.e., 300 simulations) in the last calibration iteration [24]. Validation results were quantified in the same way as calibration results using objective functions.

*2.4. Climate Model Analysis and Future Scenario Development*

The Africa CORDEX climate simulation data derived from the CMIP5 of the WCRC were used to assess climate change and variability. The Norwegian Earth System Model for global climate (NorESM-GCM) combined with the Swedish Meteorological and Hydrological Institute regional climate model (SMHI-RCM), hereafter referred to as SMHI_NCC-NorESM-M was used in this study based on MoE and UNDP (2014) recommendation. The model daily climate data was downloaded from the CORDEX domain at 0.44° resolution ($\approx$50 km) for two RCPs (4.5 and 8.5) through the Earth System Grid Federation [25] data node. Climate model time series were downloaded in netCDF3 format (Network Common Data Form). Climate data, including daily maximum near-surface air temperature (tasmax), daily minimum near-surface air temperature (tasmin), and daily precipitation (pr), were used to force the model simulation.

Three CORDEX grid points covered about 99% of AZB and were used for downscaling the climate data (Figure 3). The grid's center points were spatially connected and downscaled to the nearest rainfall/climate station. The downscaling procedure involved extracting simulation variables from the grid's data and spatially relating the data to the eight gauges so that the resulting dataset is considered representative of the location of the gauges used in a watershed model setup.

Five essential bias correction methods for precipitation and four temperature methods were used, these are linear scaling, local intensity scaling, power transformation, variance scaling, distribution mapping and delta change. All methods were compared using their skill to adjust historical RCM data to match observed data, hence providing a better match between hydrological simulations using corrected simulated climate data and observed climate data [26,27].

Two future projection scenarios were developed: (1) the RCP 4.5 scenario represents an intermediate emissions/GHGs mitigation scenario where GHGs become stable by mid-century through policy intervention, and (2) the RCP 8.5 scenario represents high emissions/no GHGs mitigation scenario where GHGs will increase until the end of the 21st century.

Future scenarios were modeled using a three-time series for the period between 2018 and 2100 (early-21st century (2018–2041 = 24 years), mid-21st century (2042–2070 = 29 years), and late-21st century (2071–2100 = 30 years) for comparisons with the current climate for AZB). The early-21st century time series was tailored to accommodate the delta-change bias-corrected data as it only provides time series up to 2041 and allows comparison with other bias correction methods for the same period. A reference scenario covering the period from 1973 to 2015 was also used to compare the current climate with future projected climates and impacts on hydrological processes.

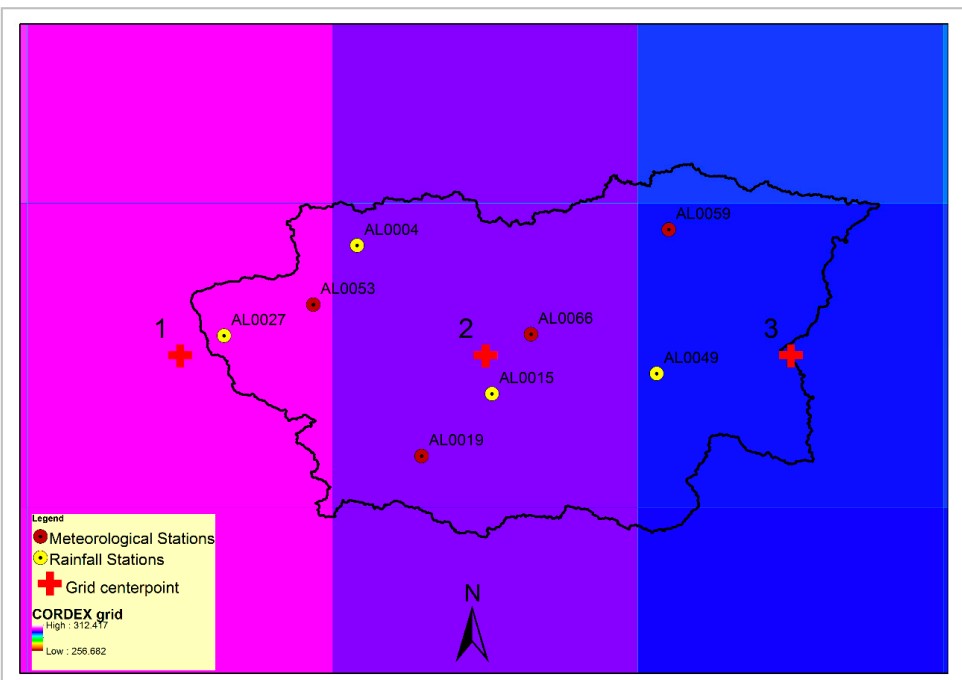

**Figure 3.** CORDEX grid center points and climate stations.

Integration of the climate and hydrologic models was achieved by forcing the calibrated SWAT model with RCM bias-corrected climate variables. At the same time, the LULC was held constant on a monthly time step for the period from 1 January 2015 to 24 November 2100, using three years as a warm-up period.

## 3. Results and Discussion

### 3.1. Reference Scenario

Manual calibration of the default SWAT run for the whole reference scenario period (1973–2015) was achieved through a sequence of parameter adjustments aiming to set the initial model in a way that properly represents the AZB surface water budget and the observed flow. [28] Revealed the importance of extensive manual calibration in a semi-arid watershed as it offers safe, systematic control of the mass balance and flow ranges while maintaining realistic watershed hydrological processes. After manual calibration, the resulting water balance components shown in Figure 4 indicate a reasonable representation of the dominant hydrologic processes and water balance ratios according to published resources [3,10,17].

The hydrograph of the observed versus simulated streamflow at the Jerash Bridge gauging station (AL0060) is shown in Figure 5. It indicates an appropriate representation of the rainfall-runoff relation of AZB. In addition, a comparison of observed versus simulated streamflow using simple statistical indicators showed satisfactory performance ($R^2$ = 0.895, RMSE: 0.538 m$^3$/s) for manual calibration ($R^2$ > 0.6, RMSE < 0.72). This is very important to have a realistic model that can be further optimized through automated calibration [18], (see Figure 6).

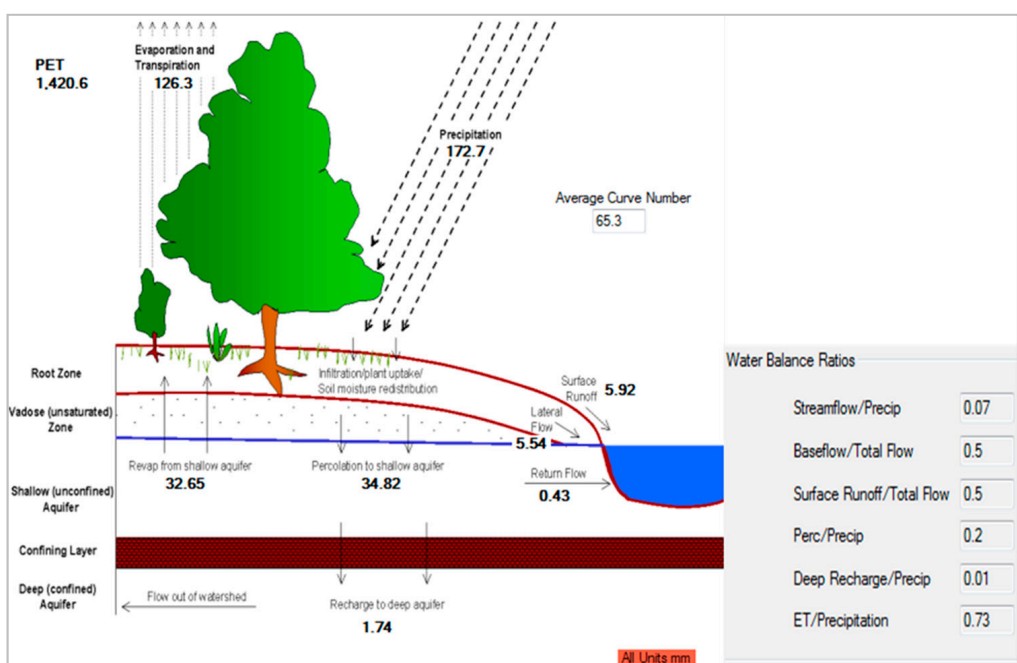

**Figure 4.** Simulated hydrology and water balance ratios for AZB for the manual calibration period (1973–2015).

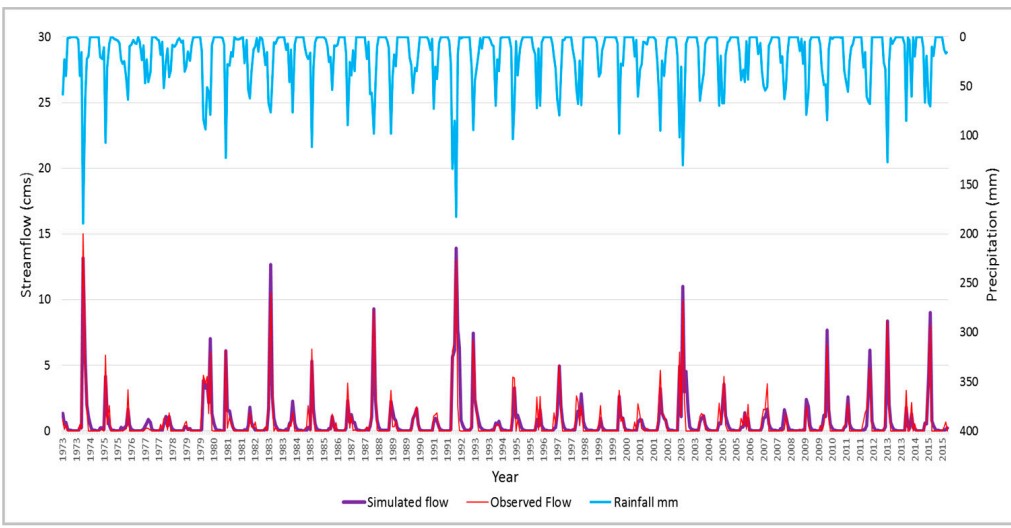

**Figure 5.** Comparison between measured and simulated monthly flood discharge for AZB at Jerash Bridge gauging station (AL0060) after manual calibration.

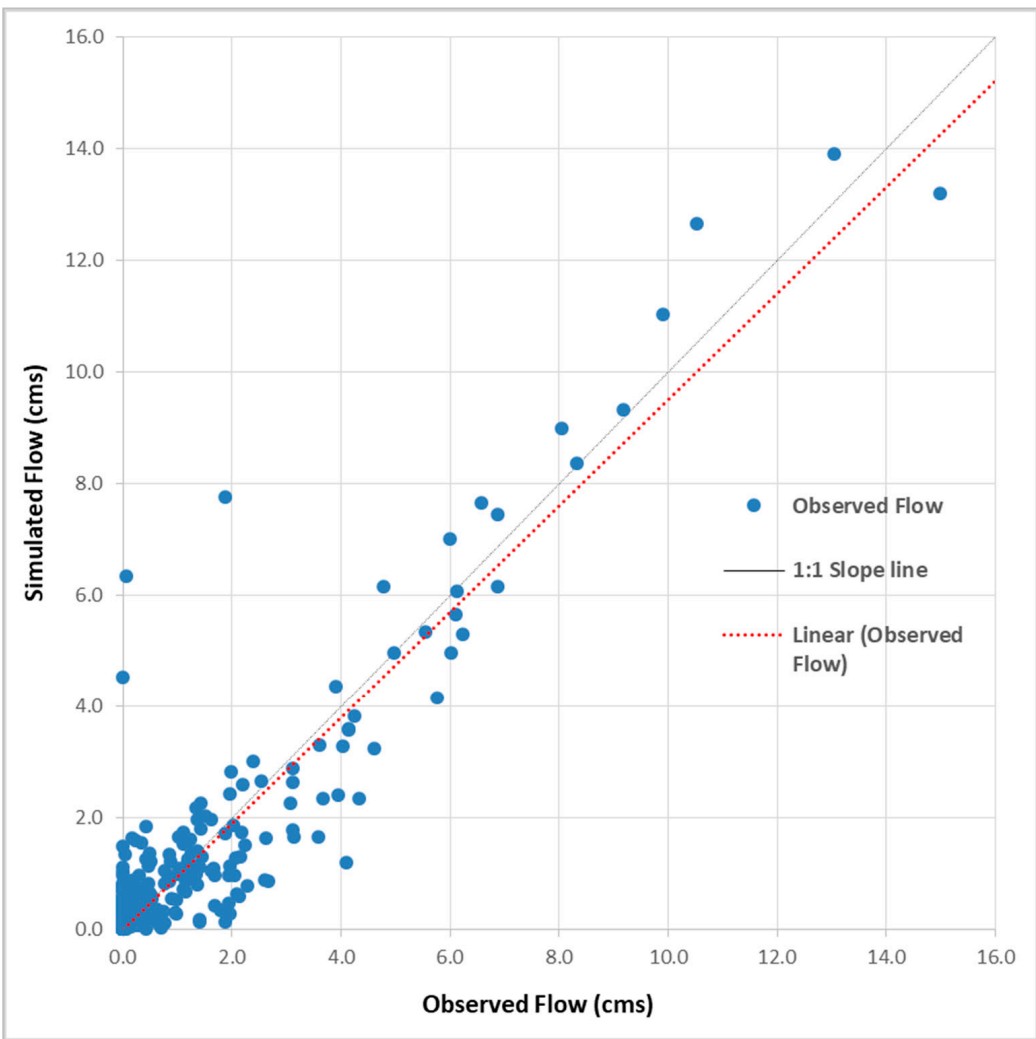

**Figure 6.** A scatter plot depicting the linear relationship between the observed and simulated monthly flood discharge for AZB at Jerash Bridge gauging station (AL0060) after manual calibration.

The manually calibrated model was further auto-calibrated using SWAT-CUP/SUFI-2 optimization algorithm. The global sensitivity analysis of the 16 commonly used parameters through the literature review resulted in the identification of the most sensitive parameters. This is important to reduce the number of parameters and avoid over-parameterization that affects the model efficiency [16,29]. The local and global parameters sensitivity analysis led to the determination of six sensitive parameters representing vital hydrological processes in AZB. Table 3 lists the most sensitive parameters and their initial and calibrated values.

**Table 3.** Sensitive parameters used for SWAT calibration for AZB.

| Parameter | Parameter Range | | Initial Value | Change Type | Initial Range | | Fitting Value | Calibrated Value |
|---|---|---|---|---|---|---|---|---|
| | Min | Max | | | Min | Max | | |
| CN2 * | 35 | 98 | 65.3 | Additive | −5 | 5 | 4.02 | 69.22 |
| SOL_K * | 0 | 2000 | 7.35 | Relative | −0.8 | 0.8 | −0.16 | 6.17 |
| SOL_AWC * | 0 | 1 | 0.106 | Relative | −0.5 | 0.5 | 0.13 | 0.12 |
| ESCO | 0 | 1 | 0.01 | Additive | 0 | 0.2 | 0.007 | 0.02 |
| SURLAG | 0.05 | 24 | 1 | Replace | 0.2 | 3 | 1.8 | 1.8 |
| GW_DELAY | 0 | 500 | 31 | Replace | 10 | 31 | 40.28 | 40.28 |

* Spatially distributed parameters: calibrated by multiplying or adding equal value to the initial value to maintain spatial variability. The initial value represents average parameter values for all subbasins.

The global sensitivity results for the calibrated values after the final optimization iteration are shown in Figure 7. The figure shows the relative parameter sensitivity based on the impact of the parameter change on the NSE objective function through multiple regression analysis (while all other parameters are changing), where the larger the t-test and the smaller the *p*-value, the more sensitive the parameter [18]. The figure shows that the CN2 is the most significant factor controlling the hydrologic balance in AZB, while the surface lag is insignificant in the final iteration. The CN2 negatively affects the parameter sensitivity (i.e., the less the value, the more it affects the objective function results). At the same time, others are positively affecting the sensitivity of the parameters.

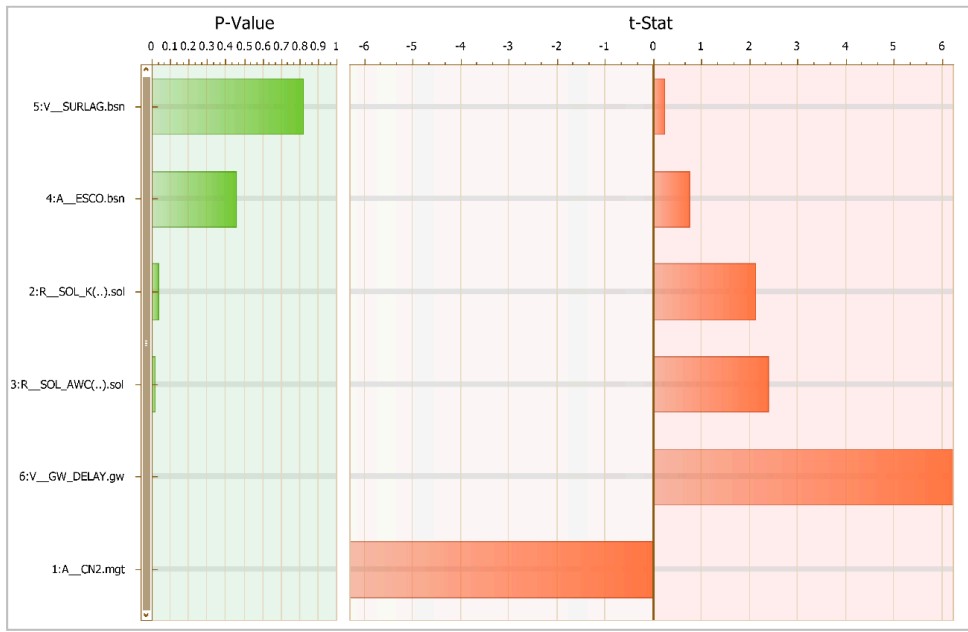

**Figure 7.** Parameterss global sensitivity analysis results for the final calibration iteration against NSE objective function.

The final set of sensitive parameters and its new best-estimated values were used to validate the model from 2003 to 2015. The objective function values for model calibration and validation periods are shown in Table 4. The overall performance based on objective functions showed a very good performance rating for both calibration and validation [30]. The difference in performance rating for calibration and validation periods is attributed to the varied hydrological conditions between these two periods that extend for 43 years, including varied wet, normal, and dry periods. In a similar study on climate change impacts on streamflow processes, very good model performance was achieved using only four years for calibration and another four for validation [20].

**Table 4.** Summary statistics of the objective functions for the calibration and validation periods.

| Objective Function | Calibration Period (1973–2002) | Validation Period (2003–2015) |
|---|---|---|
| $R^2$ | 0.9 | 0.81 |
| NSE | 0.89 | 0.86 |
| PBIAS | −6.3 | 0.9 |
| RSR | 0.34 | 0.39 |
| Mean_sim (Mean_obs) | 0.71 (0.67) | 0.64 (0.64) |
| StdDev_sim (StdDev_obs) | 1.83 (1.75) | 1.70 (1.53) |

Furthermore, SWAT usually provides poor performance in a dry period. Therefore, [31] suggested an improved seasonal calibration and simulation by separating the

time series into wet and dry periods. However, this was not incorporated into the calibration process, and the time series selection was based on the recommendation of using two-thirds of the period for calibration and the rest for validation [16]. The model had a desirably very low R-factor (0.16 for calibration and 0.14 for validation). Nonetheless, it also had a very low P-factor (>70%) for both the calibration and validation periods. This is chiefly because the SWAT-CUP requires a considerable number of time-consuming simulations without parallel processing [32], which was not possible using the free version in this research context. Even though [21] recommended a P-factor of >70% for discharge calibration, he stated that there are no specific numbers for what these factors should be. Hence, we focused more on performance ratings and model representation of the water balance components.

The SWAT model was run using the final calibrated parameters to form a final auto-calibrated model (reference model) that can be used for future scenario modeling. The model run resulted in a hydrograph that showed an appropriate representation of the rainfall-runoff relation of AZB for the calibration and validation period, with higher peak flow simulation (Figures 8 and 9). Additionally, the simulated water budget for the auto-calibrated model (Figure 10) is in general agreement with those manifested by manual calibration and figures reported by previous water balance calculations for AZB resources [3,10,17]. To conclude, despite the poor quality of input climate data and very long and meteorologically variable calibration and validation periods, it is believed that the SWAT parameterization could simulate streamflow in AZB effectively. This is evident in the good performance rating for streamflow calibration and validation, an appropriate representation of the rainfall-runoff relation in the hydrograph, and realistic water balance components and ratios.

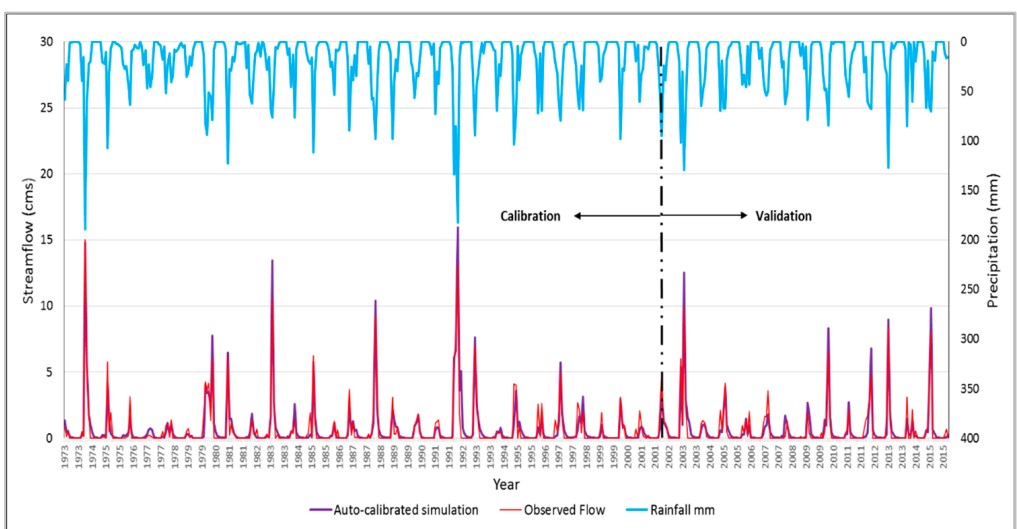

**Figure 8.** Comparison of measured and simulated monthly flood discharge for AZB at Jerash Bridge gauging station (AL0060) after auto-calibration.

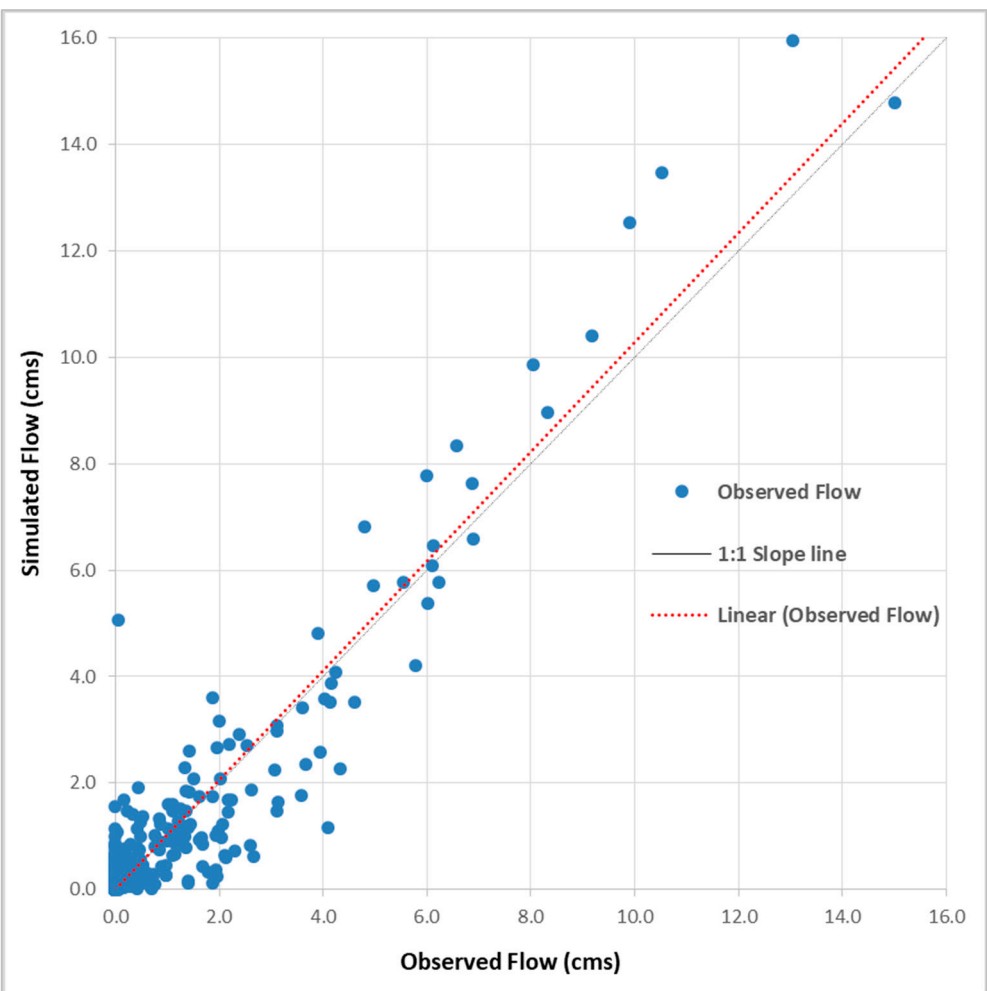

**Figure 9.** A scatter plot depicting the linear relationship between the observed and simulated monthly flood discharge for AZB at Jerash Bridge gauging station (AL0060) after auto-calibration.

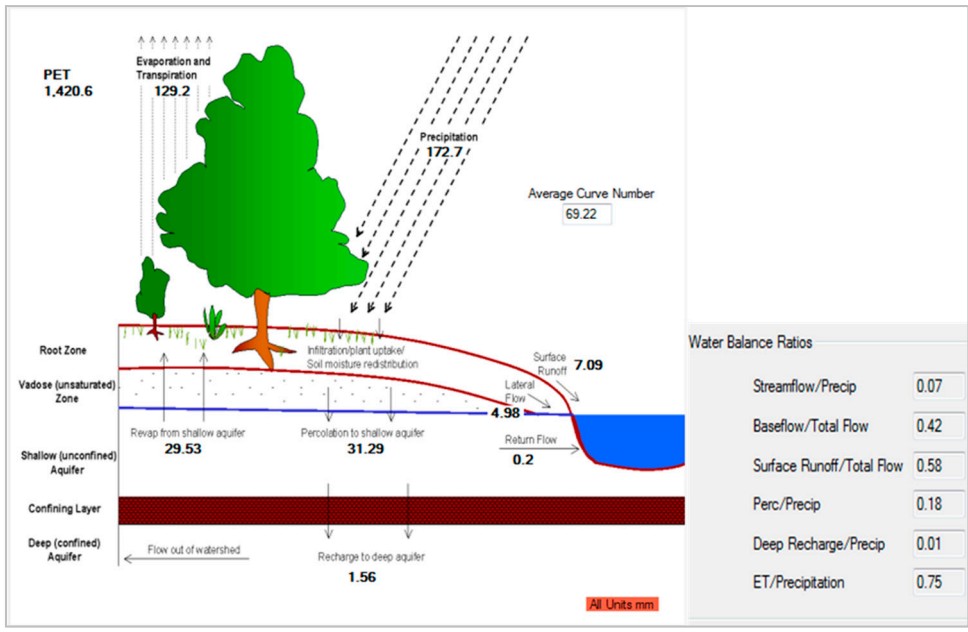

**Figure 10.** Simulated hydrology and water balance ratios for AZB for the final auto-calibrated model (1973–2015) (All units are in mm).

### 3.2. Future Scenarios

Bias correction of the climate simulation variables obtained from CORDEX RCM was conducted using essential correction methods through the CMhyd tool for the observed and bias-corrected precipitation for the control period and the default summary plots for each bias correction method for Amman Airport meteorological station (AL0019).

The observed and bias-corrected precipitation plots showed a drier trend in the annual local intensity scaling and distribution mapping methods, with the highest variation and standard deviation for distribution mapping. On the other hand, linear scaling has the smallest monthly variation and standard deviation compared to the observed data. However, it has a higher wet day probability and lowers precipitation intensity. Observed and bias-corrected maximum temperature plots showed a slight difference between all methods in terms of monthly standard deviation, 10th, and 90th percentile, except for the linear scaling method that had higher monthly standard deviation in summer and late winter months. While the minimum temperature plots show that the linear scaling method had a higher monthly standard deviation and 90th percentile, they show that it had a lower 10th percentile. Even though the variance scaling and distribution mapping are not significantly different. To simplify the selection of proper correction methods, simple testing for the goodness of fit between observed and simulated data was analyzed and presented in Table 5. Analysis of different bias correction methods skills to adjust the statistics of the respective observational climate data during the control period 1970–2005 for Amman Airport meteorological station (AL0019) showed that the best bias correction method for precipitation data is the linear scaling method, while variance scaling and distribution mapping were best methods for minimum and maximum temperature, respectively.

**Table 5.** Performance results of different bias correction methods for AL0019 station for the control period (1970–2005).

| Variable (Daily) | Bias Correction Method | RMSE (1:1 Fit Line) * | $R^2$ ** |
|---|---|---|---|
| Precipitation (mm) | Linear scaling | 4.647 | 0.004 |
| | Local intensity scaling | 4.667 | 0.0039 |
| | Distribution mapping | 4.751 | 0.0034 |
| | Power transformation | 4.866 | 0.0033 |
| Minimum Temperature (°C) | Variance scaling | 3.846 | 0.667 |
| | Distribution mapping | 3.881 | 0.662 |
| | Linear scaling | 4.235 | 0.619 |
| Maximum Temperature (°C) | Distribution mapping | 5.359 | 0.624 |
| | Variance scaling | 5.417 | 0.617 |
| | Linear scaling | 5.42 | 0.618 |

* Fitting of daily observed and bias-corrected historical data using special linear fitting (intercept = zero, and slope = 1). Residual plots showed consistent error distribution for minimum and maximum temperature and underestimated rainfall for all bias correction methods. ** Fitting of observed and bias-corrected historical data using linear fitting (*p*-value for all figures are <0.0001).

The results agree with the comprehensive review of bias correction methods presented by [27], though different performance statistics were used. The delta-change method does not adjust the RCM simulations but uses observed climate data; hence, its performance cannot be evaluated like other methods, as it gives perfect simulations by definition [27]. Based on the performance results, future hydrological modeling will combine delta-change methods for temperature and precipitation. In addition, the best-performing method for the daily minimum temperature using variance scaling and linear scaling for precipitation data will be used. The best method for daily minimum temperature was selected due to the higher change sensitivity of minimum temperature compared to a maximum temperature, as ref. [33] indicated that the annual minimum temperature has increased, while no visible

trends indicated an increase or decrease in maximum temperature in the last decade in Jordan.

The future bias-corrected climate projections up to 2100 show a clear reduction in the mean annual precipitation ranging between 10.9% and 12.8% reduction for the delta-change dataset and 14.5–15.3% reduction in the linear scaling dataset as compared to the reference period (1970–2015) (Table 6). This precipitation reduction is lower than the precipitation reduction estimated by the TNC, where a likely decrease of 20% was reported for RCP4.5 and 21% for RCP8.5 all over the country [3]. In addition, the precipitation reduction is lower than that projected using different dynamically downscaled CORDEX-RCM models, where a 30% decrease is reported for the period 2070–2100 compared to the current climate [34].

**Table 6.** Projected future scenarios mean annual values and change (Δ) relative to the reference period.

| RCP | Climate Variable | Reference Period | Delta-Change | | linear Scaling/Variance Scaling | | | | | |
|---|---|---|---|---|---|---|---|---|---|---|
| | | | EC | Δ | EC | Δ | MC | Δ | LC | Δ |
| RCP 4.5 | PCP | 221.6 | 197.4 | −24.2 | 206.8 | −14.7 | 178.9 | −42.6 | 185.5 | −36.1 |
| | TMP max | 25.5 | 27.5 | 2.1 | 26.7 | 1.3 | 27.3 | 1.9 | 28.2 | 2.7 |
| | TMP min | 11.4 | 13.5 | 2.1 | 12.8 | 1.4 | 13.4 | 2.0 | 13.9 | 2.5 |
| RCP 8.5 | PCP | 221.6 | 193.3 | −28.3 | 201.6 | −20.0 | 187.4 | −34.2 | 177.0 | −44.6 |
| | TMP max | 25.5 | 28.3 | 2.8 | 26.9 | 1.5 | 28.2 | 2.7 | 29.4 | 4.0 |
| | TMP min | 11.4 | 14.2 | 2.8 | 13.3 | 1.8 | 14.1 | 2.7 | 15.1 | 3.7 |

Notes: Precipitation is calculated based on the arithmetic mean of all stations within the basin. Early-century (EC), mid-century (MC), and late-century (LC). (PCP in mm, and TMP in °C).

Climate projections for both minimum and maximum temperatures will undergo the same increases of 2.1 °C for RCP4.5 and 2.8 °C for RCP8.5 under the delta-change dataset (Table 6). Whereas the variance scaling dataset shows the minimum temperature rises 1.4–2.5 °C for RCP4.5 and 1.8–3.7 °C for RCP8.5, while the maximum temperature rises 1.3–2.7 °C for RCP4.5 and 1.5–4 °C for RCP8.5.

The projected increase in minimum and maximum temperature for the variance scaling dataset follows that reported in the TNC, where the minimum temperature rises 2 °C for RCP4.5 and 4 °C for RCP8.5. In comparison, the maximum temperature rises 2.1 °C for RCP4.5 and 4.1 °C for RCP8.5 all over the country (MOE and UNDP, 2014). On the other hand, the delta-change dataset is not in conformity due to its shorter period. Similar mean temperature increases of 2 and 4.5 °C for RCP 4.5 and 8.5, respectively, were projected for the period 2070–2100 [34].

All future projections showed a steady temperature increase and reduced precipitation with time, except for the precipitation under the RCP4.5 scenario, where the reduction in the late century is less than mid-century. This is an inherent property of this intermediate emission pathway, also reported in the TNC [3]. Regardless of the used bias correction method, future climate scenarios showed high variability, especially in precipitation with considerable differences between RCP 4.5 and 8.5 and frequent dry events. The linear scaling method shows inconsistent future trends in both RCPs as compared to the delta-change method, where the variability between RCPs is steadier. Supplementary graphs for the projected mean annual and monthly precipitation and temperature for bias corrected future scenarios are presented in Figures S1–S8.

*3.3. Climate Change Impacts on Surface Water Resources*

3.3.1. Temporal Changes

Integration of the downscaled climate change projections from the Regional Circulation Model into the calibrated semi-distributed hydrologic model (SWAT) is used to assess potential climate change impacts on the surface water resources of AZB by studying its effects on the key water balance components. The LULC map was held constant in future scenario runs. The estimated mean annual change to the key water balance component

indicates that AZB will witness a reduction in all key water balance components for the entire modeling period; this is driven by the reduction of precipitation that ranges between about 4–21% and an increase in about 9–23% of mean annual temperatures. Figures 11 and 12 show the percent change of the key water balance component relative to the reference scenario.

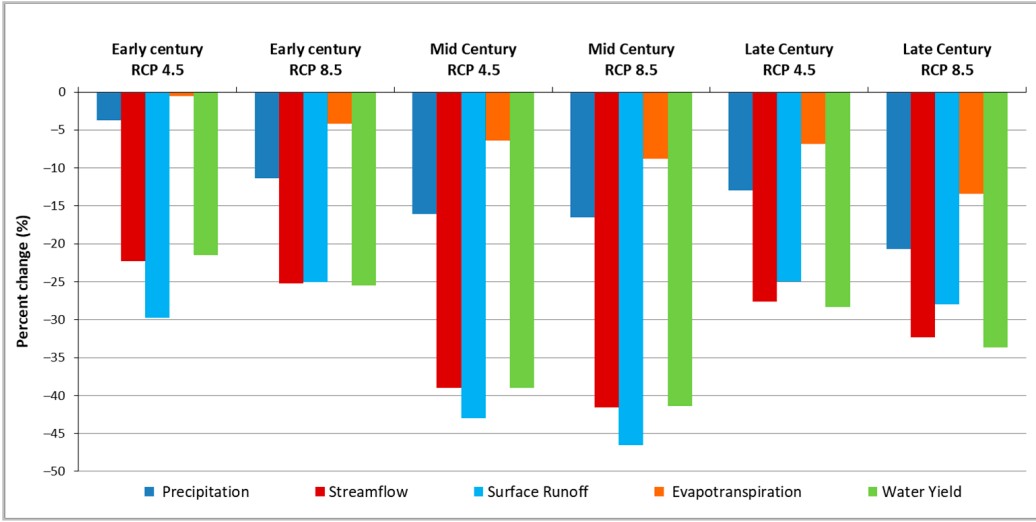

**Figure 11.** Simulated mean annual changes to key water balance components using linear/variance scaling methods.

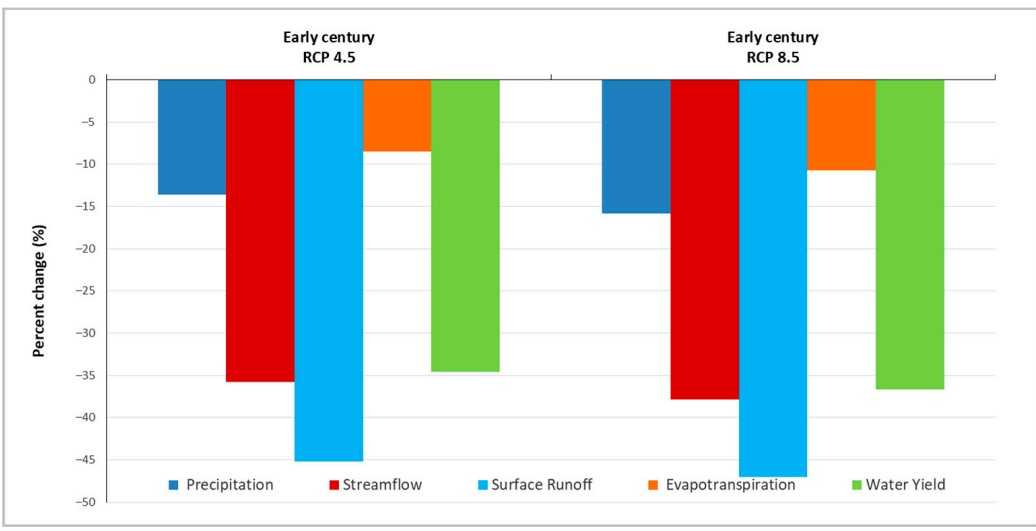

**Figure 12.** Simulated mean annual changes to key water balance components using the delta-change method.

Future scenario simulation through the SWAT model revealed the impacts on the surface water resources of AZB. Table 7 summarizes the mean annual values and simulated future change for key water balance components under different climate scenarios.

**Table 7.** Simulated changes (Δ) to mean annual water balance component of AZB under different climate change scenarios in mm (Note: Mean precipitation values represent the basin-wide average precipitation value simulated by SWAT model).

| RCP | Water Balance Component | Reference Period | Delta-Change | | Linear Scaling/Variance Scaling | | | | | |
|---|---|---|---|---|---|---|---|---|---|---|
| | | | Early-Century | Δ | Early-Century | Δ | Mid-Century | Δ | Late-Century | Δ |
| RCP 4.5 | Precipitation | 172.7 | 149.2 | −23.5 | 166.3 | −6.4 | 145.0 | −27.8 | 150.4 | −22.3 |
| | Streamflow | 12.3 | 7.9 | −4.4 | 9.5 | −2.7 | 7.5 | −4.8 | 8.9 | −3.4 |
| | Surface Runoff | 7.1 | 3.9 | −3.2 | 5.0 | −2.1 | 4.0 | −3.0 | 5.3 | −1.8 |
| | Evapotranspiration | 129.2 | 118.2 | −11.0 | 128.5 | −0.7 | 121.0 | −8.2 | 120.4 | −8.8 |
| | Water Yield | 13.8 | 9.1 | −4.8 | 10.9 | −3.0 | 8.4 | −5.4 | 9.9 | −3.9 |
| RCP 8.5 | Precipitation | 172.7 | 145.3 | −27.4 | 153.1 | −19.6 | 144.3 | −28.5 | 137.0 | −35.7 |
| | Streamflow | 12.3 | 7.6 | −4.6 | 9.2 | −3.1 | 7.2 | −5.1 | 8.3 | −4.0 |
| | Surface Runoff | 7.1 | 3.8 | −3.3 | 5.3 | −1.8 | 3.8 | −3.3 | 5.1 | −2.0 |
| | Evapotranspiration | 129.2 | 115.4 | −13.8 | 123.8 | −5.4 | 117.8 | −11.4 | 111.9 | −17.3 |
| | Water Yield | 13.8 | 8.8 | −5.1 | 10.3 | −3.5 | 8.1 | −5.7 | 9.2 | −4.7 |

### 3.3.2. Spatial Changes

The simulated spatial changes in key water balance components are presented in Figure 13. These maps represent the mean annual percent change of the key water balance component relative to the reference scenario. The spatial variability in streamflow, surface runoff, evapotranspiration, and water yield is attributed to the variability of downscaled RCM temperature and precipitation values over time and the relative location of the rainfall and meteorological stations.

(**A**) Temperature (Max)

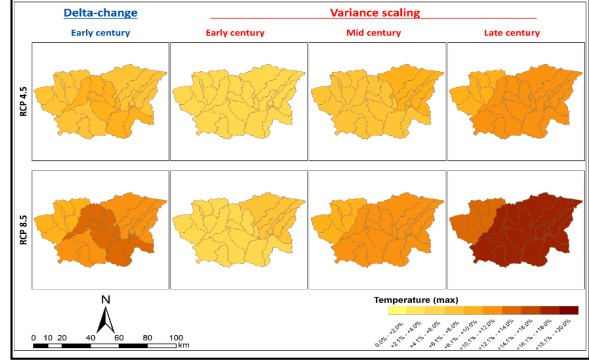

(**B**) Temperature (Min)

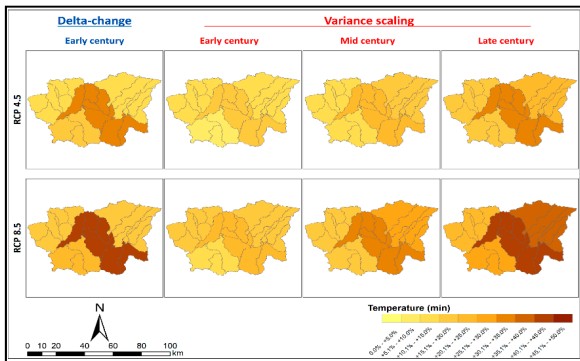

(**C**) Precipitation

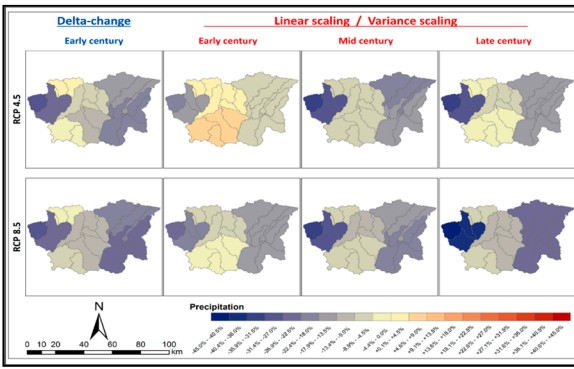

(**D**) Stream Flow

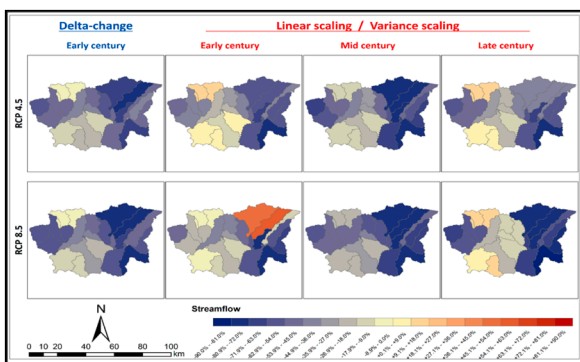

**Figure 13.** *Cont.*

(**E**) Water Yeild

(**F**) Surface Water

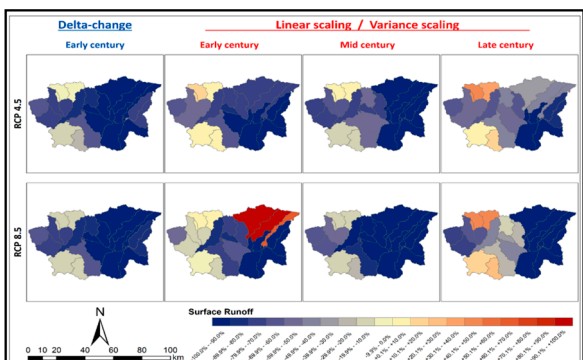

(**G**) Evapotranspiration

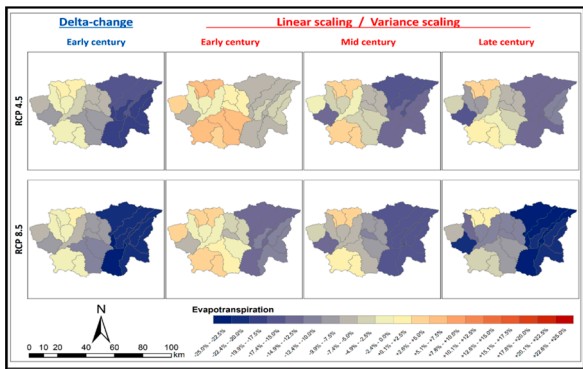

**Figure 13.** Simulated changes over AZB relative to the reference period under future scenarios.

The maximum temperature is projected to increase in the whole basin between 5–17%, where the eastern parts will have the highest rise using variance scaling and the central subbasins using delta-change under RCP8.5 (Figure 13A). The minimum temperature, however, will undergo higher increases in the range of 9–45%, where the central subbasins are projected to face greater changes (Figure 13B). The major change in minimum temperature is following the observed trend of rising minimum temperature in the central part of the basin [33].

The spatial distribution of simulated precipitation under future scenarios shows a general decline in the whole watershed (0–41%), except for a 3–7% increase in the central subbasins in the early century using linear scaling under RCP4.5, along with a very slight increase in Jerash subbasin using delta-change under RCP4.5 (Figure 13C). The overall reduction in the precipitation under the RCP4.5 scenario is more prominent in the mid-century than in the late century using the linear scaling method. The most affected areas will be the western subbasins, followed by the eastern ones.

Streamflow simulations show a general trend of decline in values, still except for subbasins with increased precipitation in the Amman and Jerash areas, and a considerable increase of about 60% in two subbasins in the northeast, which is attributed to a very low mean annual streamflow that results in high percent change caused by slight increases. Generally, the streamflow will decline in the western and eastern part of the watershed in the range of 2–82%; the drier subbasins in the east will be more influenced, which could affect the key source of perennial vegetation and grazing for local communities (Figure 13D). Water yield denotes the net amount of water that leaves the subbasin and contributes to tributary channel flow; hence, it represents streamflow minus the transmission losses to the streambed. Consequently, the simulated change in water yield shows a similar pattern to that of streamflow with only slight differences manifested by lower increases in the two abovementioned subbasins in the northeast, where the highest increase of 27% will occur in the early century under the RCP8.5 scenario (Figure 13E). Transmission losses to the

streambed are about 1.6 mm in the reference scenario and are projected to range between 0.9–1.3 mm in future scenarios.

The surface runoff shows higher spatial pattern variability than streamflow/precipitation (Figure 13F). This is due to low mean annual surface runoff in the reference and projected scenarios (≈4% of PCP), in which a very slight change in the future simulated values would have a very high percent change, depicted in the northeastern subbasins with doubled runoff amount. Generally, the surface runoff will undergo wide reductions in the whole watershed in the range of 5–100%, except for subbasins with a simulated increase of 2–100%, even so with a very slight amount of water relative to the whole water balance.

The actual simulated evapotranspiration without change in land use/land cover over time showed an increasing and decreasing pattern over the basin (Figure 13G). Precipitation reduction in the eastern subbasins under all scenarios, combined with an increase in the temperature in the mid and late century, caused reduced evapotranspiration in the eastern part of the basin. At the same time, a slight increase in precipitation and temperature in the middle subbasins in the early century caused slightly higher evapotranspiration. Jerash area and its surrounding showed increased evapotranspiration under all scenarios except for RCP8.5 using the delta-change method. The overall change in evapotranspiration will be between −24% to +7%.

Overall, the arid eastern areas of AZB are projected to become hotter and drier, with reduced water yield and ET. This will negatively impact the soil moisture content, influencing the rainfed fodder agriculture (mainly barley) and the already deteriorated rangeland vegetation, hence affecting the adaptive capacity of the local grazing community. In addition, the projected climate change in these areas that depends on flood flow rather than baseflow is likely to increase the risk of aridity and desertification.

The central areas, including the densely populated cities, will have reduced precipitation and increased ET in most of the projected scenarios, with a severe increase in the minimum temperature. This might raise the demand for agricultural and domestic water use, adding more pressure on the scarce water resources of AZB. Nonetheless, the early century projections under linear scaling will have slightly increased precipitation that can be exploited to reduce further water stresses.

The wetter, colder, hilly, and steep sloping lands in the western subbasins will have the most noticeable rainfall reduction. These areas are generally characterized by rainfed agricultural activities, especially in Baqa, Al-Balqa, and west Jerash; hence, will be vulnerable to reduced precipitation. The upstream streamflow reduction will affect the surface water availability in this area, mainly stored in KTD King Talal Dam, thus affecting the downstream-irrigated agriculture in the middle and southern parts of Jordan valley and its reliant communities. Noticeably, this important area was used as a case study in the TNC due to its established exposure to climate change, the impact of the flow of the Zarqa River, and the presence of a surface water system, stream-dependent agricultural, poor communities, and high biodiversity value [3].

## 4. Conclusions

The Amman-Zarqa Basin in Jordan was studied using a hydrologic model (SWAT) to estimate the potential impacts of climate change on its surface water resources by integration of downscaled and bias-corrected CORDEX-RCM outputs. The SWAT hydrologic modeling for AZB showed a reasonable representation of the dominant hydrologic processes and water balance ratios after manual calibration. The model was further auto-calibrated using SWAT-CUP/SUFI-2 optimization algorithm and showed a high capability of simulating streamflow with a very good performance rating for calibration and validation periods, an appropriate representation of the rainfall-runoff relation in the hydrograph, and realistic water balance components and ratios.

Statistical bias correction of downscaled CORDEX RCM using different methods indicated that the best-performing bias correction methods were linear scaling for precipitation data, along with variance scaling and distribution mapping methods for minimum and

maximum temperature, respectively. The final selection of bias correction methods was based on the best-performing method for the daily minimum temperature using variance scaling along with linear scaling for precipitation data. In addition, the widely used and robust delta-change method was also used for future scenarios despite the inability to evaluate its performance in adjusting RCM outputs.

Future climate projections showed high variability, especially in precipitation with considerable differences between RCP4.5 and 8.5, and frequent dry events. All future bias-corrected projections showed a steady increase in mean annual temperature and reduction in mean annual precipitation in successive time horizons, except for the precipitation under the RCP4.5 scenario which has a lesser reduction in the late century compared to the mid-century. The more optimistic future climate change scenario (RCP4.5) indicates a reduction in mean annual precipitation ranging between 6.6% and 19.2% for linear scaling and 10.9% for the delta-change dataset. The minimum and maximum temperatures showed equal increases of 2.1 °C for the delta-change dataset, while it ranged for the variance scaling dataset between 1.4–2.5 °C and 1.3–2.7 °C for minimum and maximum temperatures, respectively.

The temporal change to the key water balance component under the RCP4.5 scenario shows that the mid-century scenario will witness the highest impacts on all key water balance components. The simulated future reductions in key water balance components under the RCP4.5 scenario are in the range of 3.7–16.1% for precipitation, 22.3–39% for streamflow, 25–45.2% for surface runoff, 0.5–8.5% for evapotranspiration, and 21.5–39% for water yield. Spatially, the reduction in precipitation will be higher in the western subbasins followed by the eastern subbasins, while the middle subbasins will have a slight increase in precipitation in the early century causing slight increases in the other dependent water balance component in varying but insubstantial quantities. In general, the eastern arid subbasins will have the highest reduction in streamflow, surface runoff, evapotranspiration, and water yield, making them more prone to desertification.

On the other hand, the pessimistic business-as-usual scenario (RCP8.5) showed a slightly greater reduction in mean annual precipitation ranging between 9% and 20.1% for linear scaling and 12.8% for the delta-change dataset. The delta-change dataset had the same minimum and maximum temperature increase of 2.8 °C again, while it ranged for the variance scaling dataset between 1.8–3.7 °C and 1.5–4 °C for minimum and maximum temperature, respectively. The temporal change under the RCP8.5 scenario reveals that the mid-century period will witness the highest impacts on streamflow, surface runoff and water yield, while precipitation and evapotranspiration will have the highest reduction in the late century. The simulated reduction will be in the range of 11.4–20.7% for precipitation, 25.2–41.6% for streamflow, 25.1–47% for surface runoff, 4.2–13.4% for evapotranspiration, and 25.5–41.4% for water yield. All subbasins will undergo a gradual precipitation reduction under the RCP8.5 scenario with time, with a higher reduction in the western, eastern, and central subbasins progressively. Spatial variability suggests that despite some inconsistent and insubstantial increases in a few eastern subbasins, it will witness the highest reductions in the key water balance component, stressing its higher sensitivity to future changes in climate.

Overall, the projected future precipitation decrease and temperature increase, and the consequent simulation of the mean annual change to the key water balance component indicate that AZB will witness a reduction in all key water balance components for all modeled scenarios. This is in general agreement with the literature review concerning climate change's impact on water resources in similar contexts in Jordan and the Middle East region. These impacts will have negative consequences on the already scarce and stressed water resources of the basin, without adding the future non-climatic water deficit drivers. The study shows that the priority area and time horizon for proactive adaptation actions should focus on the eastern subbasins during the early century period due to their higher sensitivity to the risk of aridity and desertification.

In conclusion, the study results indicate that integrating downscaled RCM data with the SWAT model is an effective approach for assessing potential climate change impacts on the arid Amman-Zarqa surface water basin. This approach can be transferred and applied to assess hydrological responses to climate change in other mixed land-use watersheds in arid and semi-arid regions, particularly in Jordan. Moreover, the application of this approach will provide small-scale subbasin prioritization for adaptation actions in resource-poor arid regions, where climate projections are usually very coarse and reflect regional projections rather than specific subbasin conditions, hindering site-specific targeted adaptation actions.

It is recommended to take into account the main results of this study in informing 'The Action Plan for the Climate Change Policy' that will be mainstreamed in the implementation of various high-level strategies and plans. Decisively, the early century climate projections and simulated impacts on water resources overlap (if not encompass) with the timeframes of most of these plans, providing an opportunity for proactive planning and implementation of effective and timely adaptation strategies in preparation for the worse impacts on water resources projected in mid-century. Moreover, it is recommended to use the calibrated SWAT model for advancing the study of climate change impacts on groundwater resources and the water quality of the basin. This can be taken a step further by incorporating future projections of non-climatic water stress drivers and land-use changes.

**Supplementary Materials:** The following supporting information can be downloaded at: www.mdpi.com/10.3390/cli11030051/s1, Figure S1: Future mean annual precipitation scenarios using linear scaling bias correction. Figure S2: Future mean monthly precipitation scenarios using linear scaling bias correction. Figure S3: Future mean annual maximum and minimum temperature scenarios using variance scaling bias correction. Figure S4: Future mean monthly maximum and minimum temperature scenarios using variance scaling bias correction. Figure S5: Future mean annual precipitation scenarios using delta-change bias correction. Figure S6: Future mean monthly precipitation scenarios using delta-change bias correction. Figure S7: Future mean annual maximum and minimum temperature scenarios using delta-change bias correction. Figure S8: Future mean monthly maximum and minimum temperature scenarios using delta-change bias correction.

**Author Contributions:** Conceptualization, I.A.-H., M.A.-Q. and N.A.H.; methodology, I.A.-H., M.A.-Q. and N.A.H.; software, I.A.-H. and N.A.H.; validation, I.A.-H. and M.A.-Q.; formal analysis, I.A.-H. and M.A.-Q.; investigation, I.A.-H., M.A.-Q. and N.A.H.; resources, I.A.-H. and N.A.H.; data curation, I.A.-H.; writing—original draft preparation, I.A.-H., M.A.-Q. and N.A.H.; writing—review and editing, I.A.-H. and N.A.H.; visualization, I.A.-H., M.A.-Q. and N.A.H.; supervision, M.A.-Q. and N.A.H. All authors have read and agreed to the published version of the manuscript.

**Funding:** This research received no external funding.

**Data Availability Statement:** Not applicable.

**Conflicts of Interest:** The authors declare no conflict of interest.

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
