# Peer review of "Potential Impacts of Climate Change on Surface Water Resources in Arid Regions Using Downscaled Regional Circulation Model and Soil Water Assessment Tool, a Case Study of Amman-Zerqa Basin, Jordan"

_climate, doi:10.3390/cli11030051_

Round 1

Reviewer 1 Report

#1 The authors should first comprehensively compare the precipitation/temperature in the current climate before and after bias correction. And demonstrate that bias correction is necessary.

#2 What about the correlation between precipitation and temperature over the study region? Did the bias correction preserve the correlation, if any?

#3 Title: “Arid Region”.  The study is based on the basin scale. I suggest authors do not generalize the study on a global or climatology scale.

 #4 Figures 5 and 6: Show scatter plots.

#5 Figure 8 better fits before Figures 5 and 6.

#6 Table 6: What is R2? it is confusing with correlation. Too low for precipitation

 #7 Show precipitation/temperature in the future climate scenarios (RCP 4.5 and 8.5) before and after bias correction.

#8 Abstract/Conclusion section: what are the wider implications of this study?

Author Response

Comments and Suggestions for Authors

Response

1

The authors should first comprehensively compare the precipitation/temperature in the current climate before and after bias correction. And demonstrate that bias correction is necessary.

Based on the author’s experience, climate models will not perfectly fit the actual station data, so the model data was examined the earlier stages of the research work, but only the “standard procedure” to make bias correction was presented, as it is the norm for such research paper. Moreover, we didn’t include such excessive work to reduce the length of the paper

2

What about the correlation between precipitation and temperature over the study region? Did the bias correction preserve the correlation, if any?

In the study region and the whole arid surrounding regions, the correlation between precipitation and temperature is very week due to the heterogeneous nature of the precipitation spatially and temporally. So it will be very difficult to find relation between them.

Regarding the bias correction, the statistical procedures used for the bias correction uses parameterization algorithm to reproduce the observed data distribution and statistics, as the method that performs well for historic climate conditions is likely to perform well for future changed conditions

3

Title: “Arid Region”.  The study is based on the basin scale. I suggest authors do not generalize the study on a global or climatology scale.

Agreed, the title is modified to reflect a more specific case study located in ‘an arid region’, which can replicate the methodology in other regions

4

Figures 5 and 6: Show scatter plots.

Figure 5: changing the chart type will make the reader lose the representation and consistency of the rainfall and its resulted stream flow. See for example the three “Related Papers Published in MDPI Journals”:

https://www.mdpi.com/2072-4292/14/6/1314

https://www.mdpi.com/2073-4441/13/11/1560

https://www.mdpi.com/2306-5338/8/4/157

Figure 6 : is generated automatically by the SWAT-CUP and we can’t control the chart type

5

Figure 8 better fits before Figures 5 and 6.

Figure 8 shows the water balance values after the automated calibration, while figure 5 show the manual calibration results. So its better not to change the figures orders as they were presented orderly based on the research outcomes

6

Table 6: What is R2? it is confusing with correlation. Too low for precipitation

It is the coefficient of determination (R squared), used to provide simple testing for goodness of fit between observed and simulated data. Inherently, its normal to have it very low for precipitation due to the nature of “rainfall events” on spatial and temporal aspects, as arid reigns in Jordan have dispersed shower events that lasts for few hours, which leads to the fact that it has high variability of the rainfall events in the study area.

7

Show precipitation/temperature in the future climate scenarios (RCP 4.5 and 8.5) before and after bias correction.

Although this is the first step we did once we extracted the data from the climate model, we believed that it would be very lengthy to show them in the paper, however, it can be provided as supplementary materials if needed.

Moreover, the bias corrected data and statistics for future projections was produced by the CMhyd tool.

8

Abstract/Conclusion section: what are the wider implications of this study?

Amended. Elaborated  more in the conclusions

Reviewer 2 Report

I would like to thank the authors for their hard work on this manuscript. I have the following comments:

Introduction

1. You should write a new introduction to the importance of using climate models in more detail on why you choose them and how they fit into your study area, and a further literature review should be carried out in the Literature section of this article.

2. The problem of the chosen study area should be addressed in a few paragraphs of the introduction.

3. In the last paragraph of the introduction, the purpose of the work should be clearly stated, what is being done, and the basic structure of the paper should be given to the reader to better understand.

Methodology

1. There is too much literature review on SWAT principles, should add to the introduction.

2. To run future surface water results to the year 2100, it should specify whether current LULC is used or future LULC predictions are used (if present, should be returned).

So why did you make that choice and did it affect the results of your studies?

Results and discussion

1. Check the font used in table 5 7 8.

2. In part of section 3.2, which from reading thought that is the main body of this paper. Several details should be added, for example, a comparison of predicted 2100 runoff yields, in which the results of this paper show no results of future annual runoff yields. It should also show a comparison of future runoff increases or decreases by showing a graph.

3. In Section 3.3.1, when comparing the use of the RCP 4.5 and 8.5 models as shown, the final result should be discussed as to which of the two is more suitable for the selected study area.

4. Figure 4 and 8, is it necessary?

Conclusions

1. The results should be summarized in accordance with the objectives of the study completely according to the paper structure. Conclusions should be expanded to give more implications for future research and various management insights based on the study's findings, as well as limits.

Author Response

Comments and Suggestions for Authors

Response

Introduction

You should write a new introduction to the importance of using climate models in more detail on why you choose them and how they fit into your study area, and a further literature review should be carried out in the Literature section of this article.

The authors already wrote a concise introduction about the climate models (introduction, paragraph 3&4). Moreover, there are huge literature review about the climate models and hydrological models and how coupling of both models can be used for future prediction, so it will be very lengthy to do so.

The problem of the chosen study area should be addressed in a few paragraphs of the introduction.

It is addressed in the first paragraph of the section 2.1. we also added few lines to make it more clear to the reader

In the last paragraph of the introduction, the purpose of the work should be clearly stated, what is being done, and the basic structure of the paper should be given to the reader to better understand.

The purpose of the work was explicitly mentioned in the last paragraph as the research focused on estimating the potential impacts of climate change on surface water resources through integrating downscaled output from Regional Circulation Model with Soil Water Assessment Tool for Amman-Zarqa Basin.

We added a paragraph to show the basic components of the research and its structure

Methodology

There is too much literature review on SWAT principles, should add to the introduction.

We summarized the literature review to include only the most recent and relevant publications we add them to the introduction section

To run future surface water results to the year 2100, it should specify whether current LULC is used or future LULC predictions are used (if present, should be returned).

So why did you make that choice and did it affect the results of your studies?

Last paragraph in the Materials and Methods section states that the LULC was held constant on a monthly time step for the period from 01/01/2015 to 24/11/2100. So we used Fixed LULC study design.

Changing the LULC takes another work that is beyond the scope of this study

Results and discussion

Check the font used in table 5 7 8.

Amended

In part of section 3.2, which from reading thought that is the main body of this paper. Several details should be added, for example, a comparison of predicted 2011 runoff yields, in which the results of this paper show no results of future annual runoff yields. It should also show a comparison of future runoff increases or decreases by showing a graph.

The change in the main weather variables are summarised in table 7

Change in the water balance components are shown in figure 9&10

In Section 3.3.1, when comparing the use of the RCP 4.5 and 8.5 models as shown, the final result should be discussed as to which of the two is more suitable for the selected study area.

Actually both models apply as both scenarios are based on IPCC concentration pathways which cant be predicted for the study region specifically, but rather provide possible scenarios that should be considered in future risk assessment and adaptations

Figure 4 and 8, is it necessary?

Yes these figures are necessary because they show the water budget for the study area after manual and automated calibration

If we have to reduce the paper and lessen the number of figures, then we at least need to show figure 8, and consider deleting figure 4. 

Conclusions

The results should be summarized in accordance with the objectives of the study completely according to the paper structure. Conclusions should be expanded to give more implications for future research and various management insights based on the study's findings, as well as limits.

Amended. Wider implications elaborated in the conclusions

Reviewer 3 Report

Brilliant research work. Authors did a fantastic job. Very relevance for this journal.  Just a minor revision required before it can be accepted for the publication. Please see the comments in the reviewed article attached here.

Author Response

Comments and Suggestions for Authors

Response

1

Downscaled

Amended

2

outputs are

Amended

3

Before this paragraph, authors should mention other available models for similar studies and why SWAT is preferred for this study

Amended

4

Can be deleted as it is too generic information

Yes, it is generic, but necessary to show the relevance of the SWAT model for our research purpose. It also partly answer your previous question on why SWAT is preferred for this study

5

region.

Amended

6

Font size for legends in figure 2d and 2e are too small to read. please revise it

Amended & the last two maps were combined to reduce the number of maps while keeping the same content

Round 2

Reviewer 1 Report

climate-2150727 Potential Impacts of Climate Change on Surface Water Re-2 sources in Arid Regions using Downscaled Regional Circula-3 tion Model and Soil Water Assessment Tool, A Case study of 4 Amman-Zerqa Basin, Jordan.

I suggest authors respond to all comments in my previous review and modify the manuscript accordingly.

Comment #4 Figures 5 and 6: Show scatter plots. Authors should discuss whether there are systematic biases in the model simulation: underestimation/overestimation of low/high flow. Figures 5 and 6 (time-series plots) are meaningless without scatterplots

Comments  # 7 Show precipitation/temperature in the future climate scenarios (RCP 4.5 and 8.5) before and after bias correction.

 #1 in Figures 9 and 10, why precipitation is plotted as a line while all other stream flow components are shown as bar plots. water yield might be confusing with streamflow or surface runoff for readers. Should be defined in the text properly.

Author Response

Comments and Suggestions for Authors

Response

1

I suggest authors respond to all comments in my previous review and modify the manuscript accordingly.

We would like to draw the editor’s attention to the submitted responses to the first round review of the paper. We have responded to every single comment or request and would appreciate your kind consideration to our responses.

2

Comment #4 Figures 5 and 6: Show scatter plots. Authors should discuss whether there are systematic biases in the model simulation: underestimation/overestimation of low/high flow. Figures 5 and 6 (time-series plots) are meaningless without scatterplots

Scatter plots are provided in the reviewed manuscript.

There were no systematic biases in the model outputs per se, but there were some observations that cannot be generalized:

- Simulation output values are higher than observed flow data in peak rainy seasons (February).

- Simulation output values are lower than observed flow data in early rainy seasons (Nov and Dec).

- SWAT usually provide poor performance in dry period, Zhang, Chen, Yao, and Lin (2015).

3

Comments  # 7 Show precipitation/temperature in the future climate scenarios (RCP 4.5 and 8.5) before and after bias correction.

We believe that showing the future data for PCP & TMP for 83 years will not provide an added value in the main text as it will not show details for the reader, however, the data and graphs can be provided as supplementary materials if required.

Plots for bias-corrected future data are provided in a separate file to show examples of how lengthy data and graphs can be.

4

 #1 in Figures 9 and 10, why precipitation is plotted as a line while all other stream flow components are shown as bar plots. water yield might be confusing with streamflow or surface runoff for readers. Should be defined in the text properly.

The simulated precipitation volume change relative to the reference scenario is plotted in different chart types to highlight it because it is the key input to the hydrological model and driver of change that affects all the other related key water balance components in the graph. Accordingly, you can link the change in the PCP water volume with other water balance components.

A definition of water balance components can be made, but we believe that the reader of such paper should be aware of key SWAT model parameters and components that we tackled by this paper. 

If the editor believe that we have to include definitions, then we will add the following

key surface water balance component for AZB:

  • PREC: Average amount of precipitation in the watershed for the day, month or year (mm).
  • ET: Actual evapotranspiration in the watershed on the day, month or year (mm).
  • SURQ: Surface runoff in watershed for the day, month or year (mm). It represent amount generated before transmission, pothole, wetland and pond losses.
  • WYLD: Water yield to stream flow from HRUs in watershed for the day, month or year (mm). It represents the total amount of water leaving the HRU and entering main channel during the time step (WYLD = SURQ + LATQ + GWQ - Transmission Loss through tributary bed - Pond Abstraction).
  • FLOW_OUT (Streamflow): Average daily streamflow out of reach during time step (FLOW_OUT m3/s = SURQ + LATQ + GWQ).
  •  LATQ: Lateral flow (interflow) contribution to stream flow in the watershed for the day, month or year (mm).
  •  GWQ (Baseflow or return flow): Groundwater contribution to streamflow (mm). It represents the water from shallow aquifer that return to the main channel during time step.
  • PERCO LATE: Water percolation past bottom of soil profile in watershed for the day, month or year (mm).
  •  TILE Q: Drainage tile flow contribution to stream in watershed on the day, month or year (mm).
  • SW: Amount of water stored in soil profile in watershed for the day, month or year (mm).
  • PET: Potential evapotranspiration in watershed on the day, month or year (mm).